# Primary Thyroid Lymphoma: A Retrospective-Observational Study in a Single Institutional Center

**DOI:** 10.3390/medicina60030476

**Published:** 2024-03-14

**Authors:** Octavia Vita, Alis Dema, Robert Barna, Remus Cornea, Dan Brebu, Mihaela Vlad, Oana Popa, Ioana Muntean, Diana Szilagyi, Mihaela Iacob, Maria Iordache, Marioara Cornianu, Dorela Codruta Lazureanu

**Affiliations:** 1Department of Microscopic Morphology-Morphopatology, ANAPATMOL Research Center, “Victor Babes” University of Medicine and Pharmacy, 300041 Timisoara, Romaniacornea.remus@umft.ro (R.C.); muntean.liliana@umft.ro (I.M.); lazureanu.dorela@umft.ro (D.C.L.); 2Department of Pathology, “Pius Brinzeu” County Clinical Emergency Hospital, 300723 Timisoara, Romania; 3Researching Future Chirurgie 2, Department of Surgery II, “Victor Babes” University of Medicine and Pharmacy, 300041 Timisoara, Romania; 4Department of Endocrinology, Centre of Molecular Research in Nephrology and Vascular Disease, “Victor Babes” University of Medicine and Pharmacy, 300041 Timisoara, Romaniaoana.taban@umft.ro (O.P.); 5Department of Hematology, “Victor Babes” University of Medicine and Pharmacy, 300041 Timisoara, Romania

**Keywords:** primary thyroid lymphoma, Hashimoto thyroiditis, non-Hodgkin B-cell lymphomas, fine needle aspirate, survival

## Abstract

*Background and Objectives*: primary thyroid lymphoma (PTL) is a rare neoplasm, displaying a variety of histological features. It is often a challenge for pathologists to diagnose this tumor. *Materials and Methods*: this study is a retrospective analysis of clinical and pathological characteristics of a group of eleven patients (eight women and three men, mean age 68 years, range 50–80 years) diagnosed with PTL. *Results*: nine patients (81.81%) presented a tumor with progressive growth in the anterior cervical region, usually painless and accompanied by local compressive signs. Histologically, we identified six cases (55%) of diffuse large B-cell lymphoma, three cases (27%) of extranodal marginal zone lymphoma, one case (9%) of follicular lymphoma, and one case (9%) of mixed follicular-diffuse lymphoma. PTL was associated with microscopic Hashimoto autoimmune thyroiditis in ten cases (90.9%). Ten patients (90.9%) presented with localized disease (stage I-IIE). A percentage of 60% of patients survived over 5 years. We observed an overall longer survival in patients under 70 years of age. *Conclusions*: PTL represents a diagnosis that needs to be taken into account, especially in women with a history of Hashimoto autoimmune thyroiditis, presenting a cervical tumor with progressive growth. PTL is a lymphoid neoplasia with favorable outcome, with relatively long survival if it is diagnosed at younger ages.

## 1. Introduction

Primary thyroid lymphoma (PTL) is an unusual tumor, representing 1–5% of all thyroid malignant neoplasms and approximately 2.5–7% of extranodal lymphomas [1,2,3,4,5]. It is diagnosed especially in women over 50 years old, with the highest frequency being observed in the seventh and eighth decades of life [2,3].

The most important risk factor in developing PTL is the presence of Hashimoto autoimmune thyroiditis. This association is observed in over 80% of cases [6]. Patients with autoimmune thyroiditis have a 70–80 times higher risk for the development of a thyroid lymphoma than the general population [6,7,8]. Frequent clinical manifestation includes the presence of a painless tumor mass in the anterior cervical region, with rapid growth, with or without cervical lymphadenopathy, usually accompanied by symptoms of aero-digestive obstruction. Less than 20% of patients present classical systemic symptoms of B-cell lymphoma, such as fever, night sweats, and weight loss [9,10,11,12].

Most primary thyroid lymphomas are non-Hodgkin B-cell lymphomas. T-cell lymphomas, Burkitt, and Hodgkin lymphomas were rarely described [2,13]. Fine needle aspirate (FNA) is not a trustworthy diagnostic procedure to be used in cases with thyroid lymphomas, and it has a limited role, with the desirable diagnostic method being biopsy puncture guided by ultrasonography, or surgical biopsy [7,11,12,14].

The treatment of PTL is multidisciplinary (surgical, radiotherapy, chemotherapy) [9,10,13], but, because of the rarity of this disease, there is no standardized protocol. The prognosis of patients with PTL is good, but survival depends on factors such as the stage of the disease at diagnosis, the age of the patient, the histological subtype, and the elected method of treatment [2,3,12,15].

In this study, we report a group of primary thyroid lymphomas for which we analyzed the clinical, pathological, and evolutive aspects, in an attempt to contribute to a better understanding of this rare neoplasm.

## 2. Materials and Methods

The study protocol was approved by the Ethical Committee of the “Pius Brinzeu” County Clinical Emergency Hospital from Timisoara (approval number 402/26.07.2023). We carried out an observational retrospective study on a group of 11 patients with PTL, diagnosed on surgically resected specimens from the Surgery Department of the County Clinical Emergency Hospital “Pius Brinzeu” from Timisoara, Romania, between 1 January 2010 and 31 December 2019.

From the pathology reports, clinical records, operative reports, and discussions with the physicians who treated these patients, we gathered the following demographical, clinical, and pathological parameters: age and gender of the patients, clinical symptoms and signs, clinical diagnosis, method of diagnosis, localization, and dimensions (largest) of the tumor, histological type, tumor stage, association with other thyroid pathologies, treatment method, and overall survival (OS), which is defined as the duration of patient survival from the date of the pathological diagnosis and up to the time of death or the date of the last follow-up. For the analysis of the correlation between the patient’s age and OS, we divided the cases into two categories: ≤70 and >70 years old. The last access to the medical database of patients was on 1 July 2023. None of the cases received chemotherapy or radiotherapy prior to the surgical resection.

PTL cases were classified according to the World Health Organization (WHO) *Classification of Tumors of Hematopoietic and Lymphoid Tissues*, 5th edition [15]. PTL staging was carried out according to the modified Lugano classification, as follows: stage IE, the disease is localized to the thyroid tissue, with or without extension into the perithyroidal soft tissues, without involving the local and regional lymph nodes; stage IIE, the disease is localized to the thyroid and it also involves local and regional lymph nodes; and stage IV, the disease is disseminated, with nodal and/or extranodal involvement [9,15]. The patients with stage IE and IIE tumors were classified as having localized disease, while those with stage IV tumors were considered as presenting a disseminated disease. The cases diagnosed before implementing this classification were re-analyzed and staging was updated.

According to the WHO’s 5th edition of *Classification of Tumors of Hematopoietic and Lymphoid Tissues*, diffuse large B-cell lymphoma (DLBCL) was subclassified into germinal-center B-cell (GCB) and non-GCB subtypes, based on immunohistochemical (IHC) analyses [15].

Follicular lymphoma (FL) was graded based on the absolute number of centroblasts, evaluated at high magnification, observed in 10 representative neoplastic follicles, as follows: grade 1: 0–5 centroblasts/field; grade 2: 6–15 centroblasts/field; and grade 3: >15 centroblasts/microscopic field. The 5th edition of *Classification of Tumors of Hematopoietic and Lymphoid Tissues* classifies Grade 3 into two subtypes: Grade 3A, characterized by the presence of centrocytes, and Grade 3B, marked by sheets of centroblasts. Classic subtypes of follicular lymphoma include Grade 1, Grade 2, and Grade 3A FL, while Grade 3B FL is identified as follicular large B-cell lymphoma [15,16].

The sampled tissues were examined microscopically using Hematoxylin and Eosin (H&E) staining, followed by immunohistochemical analysis. The immunohistochemical method and characteristics of antibodies used in our study are presented in Table 1. Visualization was performed with the polymer system, utilizing DAB (diaminobenzidine) chromogen and hematoxylin counterstain. Moreover, for certain cases, CD30 (clone BerH2), Cyclin D1 (clone EP12), kappa (clone CH15) and lambda (clone SHL53) chains, and CD5 (clone 4C7) were assessed, all of which yielded negative results. These IHC markers have been used to rule out another type of tumor and to establish the subtype of the lymphomas. At the moment of the diagnosis, these markers were the only ones available in our department.

For the Ki-67 immunohistochemical marker, approximately 500 cells were counted using an eyepiece with a grid in a ×400 magnification, in representative areas which do not to contain residual germinal centers, hot spots of proliferation, or proliferating T cells. The Ki-67 index was calculated as the percentage of positive cells by averaging the values obtained for the two areas (count–Ki-67 index) [17]. For all other stains, we classified them as positive if more than 30% of the tumoral cells exhibited the specific antibody, whether cytoplasmic or nuclear, as specified in each stain manufacturer’s prospectus. Since there is no established requirement to report the percentage of tumoral cells for these stains, we opted to categorize them simply as positive or negative.

## 3. Results

### 3.1. Clinical and Pathological Characteristics of Patients with Primary Thyroid Lymphoma

We identified eight women (72.73%) and three men (27.27%), with a mean age at the moment of diagnosis of 68 years and a median age of 72 years (ranging from 50–80 years old). Nine patients (81.81%) presented with a prominent tumor mass situated in the anterior cervical region, with progressive growth (during 1–3 years). Some patients observed more rapid growth of the tumor during the period before diagnosis. Clinical symptoms, local or systemic, for each case are presented in Table 2.

Two patients (18.18%) had a clinical history of autoimmune thyroiditis, while another two patients (18.18%) had antithyroid antibodies (anti-thyroglobulin and anti-thyroid peroxidase). Eight patients (72.73%) were euthyroid, while three (27.27%) presented hypothyroidism.

Only one patient, with a history of autoimmune thyroiditis, was diagnosed with thyroid lymphoma after FNA, the smear highlighting extremely high cell density, consisting of apparently atypical discohesive lymphoid cells, with obvious nuclear pleomorphism. The other ten patients were diagnosed with the following after surgery: total thyroidectomy (six cases), subtotal thyroidectomy (one case), right total lobectomy (one case), and subtotal left lobectomy (three cases). The clinical findings that led patients to be referred for surgery are shown in Table 2.

We were able to appreciate the size of the thyroid glands in cases where total or partial thyroidectomy was performed; the mean size of the right thyroid lobe was 10.2 cm (range 4–17 cm) and of the left thyroid lobe was 11.01 cm (range 7.2–18 cm).

Macroscopic examination revealed the presence of solid, white, non-encapsulated, homogenous tumors, with multinodular aspect. The interface between the tumor and the surrounding thyroid parenchyma was usually ill-defined. In one case, the tumor was made up of only one nodule of 2 cm diameter, well delimited, with a grey-brown aspect.

All investigated tumors were non-Hodgkin B-cell lymphomas, CD20-positive, including six cases of diffuse large B-cell lymphoma (DLBCL), three cases of extranodal marginal zone lymphoma (E-MZL lymphoma), one case of follicular lymphoma (FL), and one case of lymphoma with mixed morphology, follicular and diffuse (Table 3). Histologically, the association between thyroid lymphoma with lymphocytic Hashimoto thyroiditis was present in 10/11 cases (90.90%). Three patients (27.27%) presented papillary carcinomas (Table 3).

Immunohistochemical profile of the tumors are presented in Table 4.

For ten patients (90.9%) the tumor was localized (seven patients had stage IE and two patients had stage IIE), and for one patient (9.1%), the tumor was diagnosed as stage IV (Table 3).

### 3.2. Clinical and Pathological Aspects in DLBCL

DLBCL was the most frequent histological subtype of primary thyroid lymphoma that we encountered during this study, being diagnosed in six patients (54.54%): four women and two men, mean age of 68.5. Five of these patients presented with a tumor mass in the anterior cervical region, with compressive symptoms. In two patients, systemic lymphoma-associated signs were also present.

Microscopically, DLBCL was made up of large, atypical lymphoid cells, discohesive, with moderate or high pleomorphism (Figure 1a), frequent apoptotic bodies, and tumor necrosis (Figure 1b). Lymphoepithelial lesions (Figure 1c) were observed in four cases. All cases showed tumor cells positive for CD20 immunostain (Figure 1d). Out of the six patients diagnosed with DLBCL, three cases were classified as GCB subtype and the other three were non-GCB subtype.

We observed microscopic lesions of Hashimoto’s lymphocytic thyroiditis in all patients, with one patient also having a papillary microcarcinoma. Three patients (50%) were diagnosed with stage IE, two patients (33.33%) had stage IIE tumors, and one patient (16.7%) was diagnosed with stage IV.

### 3.3. Clinical and Pathological Aspects of E-MZL Lymphoma

E-MZL lymphoma was diagnosed in three patients (27.27%), two women and one male, mean age 69.9 years. Clinically, the patients presented a prominent mass in the anterior cervical region, but obstructive symptoms were inconsistent.

Histopathological examination revealed the replacement of the thyroid parenchyma by a lymphoid tumor proliferation with a diffuse growth pattern, made up of small tumor cells, relatively monomorphic, and “centrocit-like” (Figure 2a). Lymphoepithelial lesions were observed, defined as aggregates with at least three lymphoid cells, localized in the epithelium of thyroid follicles and/or endoluminal, with distortion or destruction of the follicular epithelium. Lymphoepithelial lesions varied in quantity and quality, being found under two forms. The first and more frequent one was characterized by the presence of neoplastic cells arranged in nests with variable sizes, inside and in between the thyroid follicular epithelium (Figure 2b,c). The second and more unusual form was defined by round masses of neoplastic cells of variable dimensions that filled and distended the intrafollicular space, those being lymphoepithelial lesions with a “ball-like” appearance. IHC, neoplastic lymphoid cells were CD20-positive (Figure 2d).

In two patients, the lymphoma developed on a lesion of Hashimoto thyroiditis. In the third patient, lymphoid proliferation affected the entire thyroid gland, without any evidence of remaining thyroid non-tumor parenchyma. In one patient, we noted the association between lymphoma with papillary carcinoma. All patients presented with localized disease, being diagnosed with stage IE tumors.

### 3.4. Clinical and Pathological Aspects in FL and Mixed Follicular-Diffuse Lymphoma

FL was diagnosed in one female patient aged 54 years old, with a history of euthyroid goitre, and a high level of anti-thyroglobulin antibodies. At ultrasonography, the tumor was a well-delimited nodular lesion. Microscopic examination showed the presence of a dense, abundant, atypical lymphoid infiltrate made up of crowded neoplastic lymphoid follicles with large, irregular germinative centers, sometimes confluent, with centroblasts and centrocytes randomly distributed; lymphoepithelial lesions were present and macrophages with tangible bodies were absent. Thyroid parenchyma surrounding the lymphoid proliferation had lesions of Hashimoto thyroiditis and included a papillary microcarcinoma. IHC analysis showed centrofollicular neoplastic cells that were positive for CD20, CD79a, Bcl 6, CD10, and expressed limited positivity for Bcl 2. The tumor was confined to the thyroid gland (stage IE) and was graded as classic FL (grade G3A).

One lymphoma with mixed morphology, follicular and diffuse, was observed in a female patient aged 74 years old, with a history of euthyroid goitre. The tumor extended diffusely throughout the entire thyroid gland, which was very much enlarged and caused compressive symptoms. Microscopically, in the left thyroid lobe, we identified an atypical lymphoid proliferation, with a diffuse pattern of growth, CD20-positive, and histological features corresponding to DLBCL. In the right thyroid lobe, we described a lymphoid tumor proliferation as having a follicular pattern (Figure 3a) with numerous lymphoepithelial lesions, a lot of them “ball-like” (Figure 3b), and histological aspects compatible with a classic FL (grade 3A). The centrofollicular neoplastic cells displayed strong CD20 immunopositivity (Figure 3c,d), while CD10, Bcl-2, and Bcl-6 exhibited negative expression. This lymphoma was limited to the thyroid gland (stage IE).

### 3.5. Treatment and Survival Data

We gathered data regarding post-surgical oncological treatments for six patients. They received the R–CHOP regimen: an anti-CD20 monoclonal antibody (rituximab) and chemotherapy: cyclophosphamide, doxorubicin, vincristine, and prednisone (Table 3).

Survival data for a 3-year period following diagnosis were available for all patients, with 73% still alive. Over a 5-year period, information was available only for ten out of eleven patients, with 60% of them still alive. At the time of data collection and during this study, one patient had only been diagnosed for 4 years, which did not allow inclusion in the analysis for the 5-year survival rate. Detailed information regarding the survival of individual patients is provided in Table 3.

In patients diagnosed with E-MZL lymphomas, we noted the following course of disease: one patient (13.3%) survived for 9 years, one patient (33.3%) survived less than a year, and another patient (33.3%) died 2 years after diagnosis due to an aortic aneurism rupture.

Four out of six (66.7%) patients with DLBCL were still alive 5 years after their diagnosis, and all of these patients had localized disease. Moreover, one of these patients survived for 10 years, while another was still alive 13 years after diagnosis.

The patients with pure FL were still alive 13 years after diagnosis; the patient with mixed lymphoma—follicular and DLBCL—was alive 4 years after diagnosis.

We classified patients into two categories: those aged 70 or younger and those older than 70. The mean age for the first group was 59.6 years, with a median age of 59 years, while the second group had a mean age of 75 years and a median age of 74.5 years. Among the patients in the first group, four out of five (80%) survived more than 5 years, while in the older group, only two out of six (40%) patients survived more than 5 years.

## 4. Discussion

PTL continues to cause a lot of diagnostic and therapeutic dilemmas. The progress made in recognizing its histological features and the IHC techniques that are currently available has contributed to enhancing the diagnosis of PTL.

The majority of PTLs are found in women, with the ratio between females and males varying from 2:1–8:1 in various studies [1,4,5]. The predominance of PTL in women is worth mentioning, especially when compared to the prevalence of lymphomas in general, the latter being more frequently encountered in men in almost all subtypes [18]. The usual age at the moment of diagnosis is 50–80 years old, with a maximum during the seventh and eighth decade of life. PTL is rarely diagnosed under 40 years of age [1,2,3]. Similar to data from the literature, in our study the majority of PTLs were diagnosed in women, with a ratio of women to men of 2.66:1. All patients were ≥ 50 years old, with the mean age at the moment of diagnosis being approximately equal in both sexes. Most patients from our study were diagnosed in the eighth decade of life.

The most frequent clinical presentation in PTL is the progressive enlargement of the thyroid, in some cases during a short time, with or without cervical lymphadenopathy, accompanied by compressive signs on the surrounding structures: dysphagia, dyspnea, dysphonia, or paresis of the vocal cords. These clinical aspects were also present in our study, the majority of patients presented with a cervical tumor mass of variable sizes; in more than half of the patients, compressive signs were reported, especially on the air and digestive tracts. Systemic symptoms were observed in only two patients, both being diagnosed with DLBCL. Additionally, systemic symptoms rarely appear in extranodal lymphomas [13,19].

PTL is almost always associated with Hashimoto autoimmune thyroiditis, the latter representing a major risk factor in the development of lymphoma. The relationship between thyroid lymphoma and autoimmune disease with chronic antigenic stimulation and accumulation of lymphoid tissue is well known. However, only 0.5% of patients with Hashimoto thyroiditis will develop a lymphoma, after 20–30 years of progress of the inflammatory process [3,5,6,8,12,20,21]. In our study, pathological examination showed the association between PTL and autoimmune thyroiditis in almost all cases. It is possible that in the only case that did not show the presence of lymphocytic thyroiditis, the lymphoid proliferation may have completely obliterated the residual thyroid tissue.

Even if the high risk of lymphoma in patients with autoimmune thyroiditis is well known, the diagnosis of this neoplasia is still difficult to achieve because the clinical picture, serological analysis, thyroid functional tests, and ultrasonography are not specific for detecting lymphoproliferative infiltrate. In our study, the diagnosis of thyroid lymphoma was clinically suspected in only one patient.

FNA has an important role in the diagnosis of thyroid nodules, but it has a limited role in the diagnosis of PTL. This is because, in some cases, it is difficult to differentiate a lymphoma from a lymphocytic thyroiditis or an anaplastic thyroid carcinoma. However, FNA is a potential diagnostic method for PTL when using a cytological examination made by a skilled pathologist, combined with an IHC analysis; regardless, biopsy is essential and should remain the standard method in diagnosing this neoplasia [9,12,14]. In our study, the diagnosis of PTL was made by FNA in only one patient with a history of autoimmune thyroiditis. In all the other patients, the diagnosis was made by surgical biopsy.

Histologically, the great majority of PTLs are B-cell non-Hodgkin lymphomas, and the most frequent types are DLBCL and E-MZL lymphomas. Thyroid FL is unusual [2,9,12], while Hodgkin lymphoma, Burkitt lymphoma, and lymphocytic lymphoma are very rare [2,13]. Only a few cases of T-cell lymphomas were reported [22]. In our study, all PTLs were B-cell lymphomas, CD20-positive. Similar to data from the literature, the most frequent histological type was DLBCL, followed by E-MZL lymphoma, and FL.

The association between FL and DLBCL was rarely described in the thyroid [23]. The case of mixed lymphoma, follicular and diffuse, presented in this study is very particular, with one thyroid lobe presenting a tumor with FL morphology and the other lobe presenting a tumor with DLBCL morphology.

The transformation of an FL into a diffuse lymphoma is possible, most frequently into a diffuse lymphoma with large B cells [24]. Some authors suggest that there is a time interval of at least 6 months between the diagnosis of the initial FL and the diffuse lymphoma with large cells to determine an unequivocal transformation [25,26]. Another problem with transformed FLs is their definition, which is based on the histological evidence of DLBCL in patients with a history of FL. Histological transformation, which remains a rare event in the natural evolution of follicular lymphoma, varies between 4–27% [26].

Studies show that PTLs are diagnosed in incipient stages [1,2,3,4,5,7,12,27]. After analyzing the Surveillance Epidemiology and End Results database over 32 years for 1408 patients with PTL, Graff-Baker et al. observed that 88% of patients were diagnosed with stage I-IIE [2]. Moreover, in another study, Chai et al. reported that 92.1% of patients were diagnosed with stage I-IIE [7]. In a study of 2215 patients with PTL diagnosed between 1983 and 2015, Zhu et al. observed that 86.7% of patients were also diagnosed in stages I–IIE [27].

Generally, the evolution of PTL is good, with a high 5-year survival rate. A worse prognosis is associated with higher age (over 80 years old), advanced stage of the disease, lack of treatment, and follicular or diffuse large cell histological subtypes [1,2,3]. In a study on 108 patients with PTL, Derringer et al. reported a 5-year survival of 79% [3]. Graff-Baker et al. recorded a 5-year disease-specific survival, depending on histological subtype, as follows: 96% for E-MZL lymphoma, 75% for DLBCL, and 87% for FL [2].

In our study, similar to data from the literature, the great majority (90.9%) of PTL were localized tumors, classified as stage I-IIE. Patients with E-MZL lymphoma and follicular lymphoma were diagnosed with stage I. A percentage of 50% of DLBCL was discovered in a more advanced stage of the disease (stages IIE or IV).

In terms of post-diagnosis survival, 60% survived beyond 5 years, and 40% lived over 9 years. Those diagnosed at or below 70 years of age exhibited longer survival, probably due to a multitude of factors such as their robust immune system, tumor characteristics, aggressive treatment approach, early detection, higher treatment adherence, and better response to therapy.

However, due to the small patient sample and inadequate follow-up data, our ability to conduct thorough statistical analysis to evaluate potential correlations between lymphoma types and patient survival rates was constrained.

The surgical treatment of PTL is controversial. Although some authors questioned the necessity of thyroidectomy [9,28], in some institutions the therapeutic strategy implies the resection of the affected gland. Surgical excision must be followed by radiotherapy, including regional lymph nodes, combined with chemotherapy R-CHOP [9]. Applying chemotherapy in patients with localized disease is still controversial, but an increasing tendency towards its use was observed [21]. In our study, all patients were treated with surgery; in six patients, we gathered information about receiving chemotherapeutic treatment after surgery, although most of them presented with localized disease.

### Limits of the Study

The limitations of the study include its retrospective nature, potentially leading to omitted data. Additionally, the research was conducted on a limited number of patients, with incomplete information available on the therapy administered post-surgical intervention for some individuals and with non-standardized follow-up of the patients. Future studies necessitate a more comprehensive approach, involving larger patient cohorts with standardized and complete clinical data and follow-up, to establish the role of certain clinical and pathological parameters as prognostic factors in the progression of patients with primary thyroid lymphoma. Furthermore, the utilization of molecular testing in upcoming research would be beneficial in determining the mutations involved in this condition and whether they differ based on histological types.

## 5. Conclusions

Primary thyroid lymphoma is a rare neoplasia, but this tumor must be taken into account, especially in women with a history of autoimmune lymphocytic thyroiditis presenting with a rapidly growing tumor mass in the anterior cervical region, accompanied by local compressive signs. It is a lymphoid neoplasia with a favorable outcome and a relatively long survival if it is diagnosed at younger ages, most likely due to the overall status of the patient. These findings need to be confirmed through large-scale multicenter studies.

While our study may not have revealed new insights into the pathological development of these tumors or treatment approaches distinct from previous research, it contributes to a deeper understanding of this disease within the medical literature. The accumulation of data from similar studies provides a foundation for future investigations. We emphasize that a pertinent statistical analysis can and should be undertaken when more cases are reported in the literature, thereby advancing our comprehension and management of the disease.

## Figures and Tables

**Figure 1 medicina-60-00476-f001:**
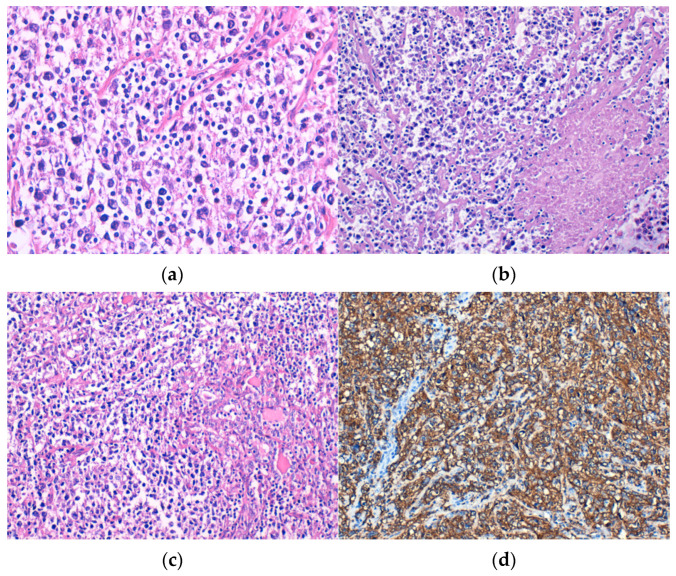
Diffuse large B-cell lymphoma (DLBCL) consisting of discohesive atypical lymphoid cells with (**a**) moderate pleomorphism, H&E staining; (**b**) tumor necrosis, H&E staining; and (**c**) lymphoepithelial lesions, H&E staining. Immunohistochemical positivity of tumor cells for (**d**) CD20. Original magnification ×400 (**a**) and ×200 (**b**–**d**).

**Figure 2 medicina-60-00476-f002:**
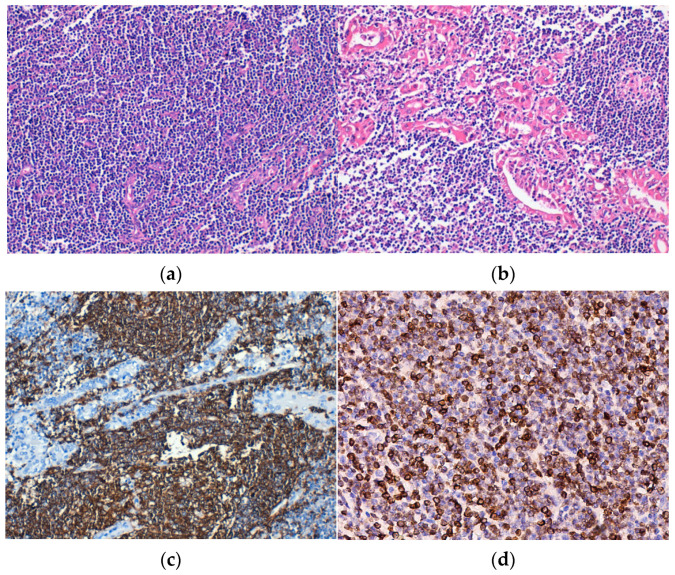
Extranodal marginal zone lymphoma (E-MZL). (**a**) Tumor consisting of small, monomorphic lymphoid cells that replace the follicular architecture, H&E staining; (**b**) Lymphoepithelial lesions, H&E staining. Immunohistochemical positivity of tumor cells for (**c**) CD20 and (**d**) BCL2. Original magnification ×200 (**a**–**d**).

**Figure 3 medicina-60-00476-f003:**
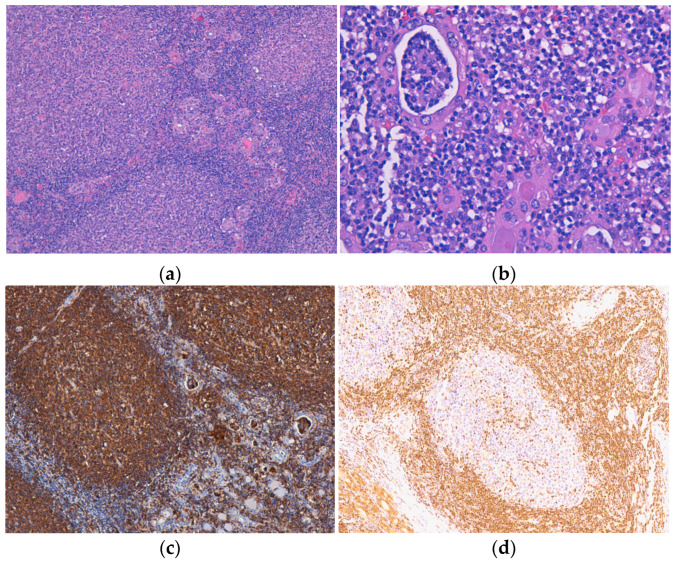
Follicular lymphoma. (**a**) Atypical lymphoid infiltrate is organized in the form of neoplastic follicles with irregular and enlarged germinal centers, H&E staining; (**b**) Lymphoepithelial lesions with a “ball-like” aspect, H&E staining. Immunohistochemical positivity of tumor cells for (**c**) CD20 and with limited (focal) positivity for (**d**) BCL2. Original magnification ×100 (**a**,**c**), ×200 (**d**) and ×400 (**b**).

**Table 1 medicina-60-00476-t001:** Characteristics of antibodies and the immunohistochemical method used in our study.

Antibody (Manufacturer)	Clone	RTU/Dilution	Antigen Retrieval	Primary Antibody Incubation Time	Visualization System
CD20 (Leica, Deer Park, IL, USA)	L26	RTU	Epitope Retrieval Solution 1 pH 6	15′	BOND Polymer Refine Detection
BCL2 (Leica, Deer Park, IL, USA)	124	RTU	Epitope Retrieval Solution 1 pH 6	15′	BOND Polymer Refine Detection
BCL6 (Leica, Deer Park, IL, USA)	PG-B6p	RTU	Epitope Retrieval Solution 1 pH 6	15′	BOND Polymer Refine Detection
CD10 (Leica, Deer Park, IL, USA)	56C	RTU	Epitope Retrieval Solution 1 pH 6	15′	BOND Polymer Refine Detection
MUM1 (Dako-Agilent, Santa Clara, CA, USA)	MUM1p	RTU	Epitope Retrieval Solution 1 pH 6	15′	BOND Polymer Refine Detection
Ki-67 (Leica, Deer Park, IL, USA)	MIB-1	RTU	Epitope Retrieval Solution 1 pH 6	15′	BOND Polymer Refine Detection
CD79a (Leica, Deer Park, IL, USA)	JCB117	RTU	Epitope Retrieval Solution 1 pH 6	15′	BOND Polymer Refine Detection

RTU = ready to use.

**Table 2 medicina-60-00476-t002:** Clinicopathological characteristics of the patients with primary thyroid lymphoma.

Case Sex, Age (Years)	Clinical Diagnosis	Diagnostic Method	Local Signs	Clinical Symptoms	Hormonal Status	History of Autoimmune Thyroiditis
F, 59 years old	No clinical diagnosis	subtotal left lobectomy	NA	NA	NA	NA
F, 65 years old	No clinical diagnosis	subtotal left lobectomy	NA	NA	NA	NA
F, 54 years old	Cold thyroid nodule	subtotal left lobectomy	cervical neck mass	no obstructive symptoms	euthyroidism, antithyroglobulin antibodies	NA
M, 80 years old	Unspecified tumor	total right lobectomy	cervical neck mass	no obstructive symptoms	NA	NA
F, 73 years old	Lymphoma	FNA followed by total thyroidectomy	cervical neck mass	dysphagia, dyspnea, cough, dysphonia	hypothyroidism	Yes
F, 70 years old	Nodular goiter	total thyroidectomy	cervical neck mass	dysphagia, dyspnea	NA	NA
F, 75 years old	Nodular goiter	total thyroidectomy	cervical neck mass	dysphagia, dyspnea	hypothyroidism	Yes
M, 50 years old	Nodular goiter	total thyroidectomy	cervical neck mass	dysphagia, dyspnea, local pain, systemic symptoms	hypothyroidism, antithyroglobulin and antithyroid peroxidase antibodies	NA
F, 76 years old	Anaplastic carcinoma	subtotal thyroidectomy	cervical neck mass	dysphagia, dysphonia, systemic symptoms	NA	NA
M, 72 years old	Nodular goiter	total thyroidectomy	cervical neck mass	dysphagia, dyspnea, dysphonia	euthyroidism	NA
F, 74 years old	Nodular goiter	total thyroidectomy	cervical neck mass	dysphagia, dyspnea	euthyroidism	NA

F: female, M: male, NA: not available.

**Table 3 medicina-60-00476-t003:** Pathological characteristics, treatment, and overall survival of the patients with primary thyroid lymphoma.

Case Sex, Age (Years)	Histologic Type	Stage	Sites of Lymphoma Involvement Associated with Thyroid Lymphoma	Other Malignancies	Associated Hashimoto Thyroiditis	Treatment after Surgery	Overall Survival
F, 59 years old	E-MZL	IE	absent	papillary carcinoma	present	R-CHOP	Died of aortic aneurysm rupture 2 years later
F, 65 years old	DLBCL non-GCB	IE	absent	absent	present	NA	Alive—disease free after 13 years
F, 54 years old	Follicular lymphoma	IE	absent	papillary microcarcinoma	present	R–CHOP	Alive—disease free after 13 years
M, 80 years old	E-MZL	IE	extracapsular extension and in adipose tissue	absent	present	NA	Died, <1 year later
F, 73 years old	DLBCL GCB	IE	absent	absent	present	NA	Died, 10 years later
F, 70 years old	E-MZL	IE	extracapsular extension and in adipose tissue	absent	absent	NA	Died, 9 years later
F, 75 years old	DLBCL GCB	IE	extracapsular extension and in adipose tissue; lympho-vascular invasion	absent	present	R–CHOP	Died, 1 year and 2 months later
M, 50 years old	DLBCL non-GCB	IIE	superior mediastinal lymph nodes metastases	papillary microcarcinoma	present	R–CHOP	Alive—disease free after 6 years
F, 76 years old	DLBCL non-GCB	IV	latero-cervical, superior mediastinal lymph nodes and ileum metastases	absent	present	R–CHOP	Died, 4 years later
M, 72 years old	DLBCL GCB	IIE	right recurrent lymph nodes metastases	absent	present	NA	Alive—disease free after 5 years
* F, 74 years old	Follicular and diffuse (DLBCL) lymphoma	IE	absent	absent	present	R–CHOP	Alive—disease free after 4 years

F: female; M: male; E-MZL: extranodal marginal zone lymphoma; DLBCL: diffuse large B-cell lymphoma; R–CHOP: rituximab, cyclophosphamide, doxorubicin, vincristine; NA: not available. * This patient had been monitored for only 4 years at the time of the study.

**Table 4 medicina-60-00476-t004:** Immunohistochemical profile of the tumors.

Case Sex, Age (years)	Histologic Type	Immunohistochemistry Markers						
		CD20	BCL2	BCL6	CD10	MUM1	Ki-67	CD79a
F, 59 years old	E-MZL	**+**	**+**	NA	NA	NA	NA	NA
F, 65 years old	DLBCL non-GCB	**+**	NA	**-**	**-**	**+** (focal)	NA	NA
F, 54 years old	FL	**+**	**+** (focal)	**+**	**+**	NA	NA	**+**
M, 80 years old	E-MZL	**+**	NA	NA	NA	NA	NA	NA
F, 73 years old	DLBCL GCB	**+**	NA	**-**	**+**	**-**	NA	NA
F, 70 years old	E-MZL	**+**	**+**	**-**	+ (focal)	NA	**30%**	NA
F, 75 years old	DLBCL GCB	**+**	NA	**-**	**+**	**-**	NA	NA
M, 50 years old	DLBCL non-GCB	**+**	**+**	**+**	**-**	**+**	**75%**	NA
F, 76 years old	DLBCL non-GCB	**+**	NA	**-**	**-**	**+** (focal)	**60%**	NA
M, 72 years old	DLBCL GCB	**+**	**-**	**+**	**-**	NA	NA	NA
F, 74 years old	FL and diffuse (DLBCL) lymphoma	**+**	**-**	**-**	**-**	NA	**75%**	NA

E-MZL: extranodal marginal zone lymphoma; DLBCL: diffuse large B-cell lymphoma; GCB: germinal-center B-cell subtype; non-GCB: non germinal-center B-cell subtype; FL: Follicular lymphoma; NA: not available; Tumor cells expressing the antibody in ≥30% of cells are considered positive (+), while those exhibiting the antibody in less than 30% of cells are considered negative (-).

## Data Availability

All data generated or analyzed during this study are included in this published article.

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
