# Peer review of "Primary Thyroid Lymphoma: A Retrospective-Observational Study in a Single Institutional Center"

_medicina, 2024, doi:10.3390/medicina60030476_

Round 1
Reviewer 1 Report
Comments and Suggestions for Authors
This study aimed to evaluate the clinical and pathological characteristics of a small number of patients diagnosed with primary thyroid lymphoma. Overall the study is well conducted and detailed. Probably, due to the small number of cases, specific analyses are lacking, such as logistic regression analysis to evaluate the associations between risk factors and onset of the disease, and Cox regression analysis to evaluate the risk of mortality in relation to selected variables. In the Conclusion section the authors should add that the observed results need to be confirmed in larger multicenter studies.
Specific comments:
Line 88. DBCL should be provided with the full name on the first use.
Author Response
We thank you for your consideration!
”In the Conclusion section the authors should add that the observed results need to be confirmed in larger multicenter studies.”
Response: We appreciate your suggestion. We modified the conclusion accordingly and we added a new paragraph (lines 446-458).
Primary thyroid lymphoma is a rare neoplasia, but this tumour must be taken into account, especially in women with a history of autoimmune lymphocytic thyroiditis presenting with a rapidly growing tumour mass in the anterior cervical region, accompanied by local compressive signs. It is a lymphoid neoplasia with a favourable outcome, with a relatively long survival if it is diagnosed at younger ages. These findings need to be confirmed through large-scale multicenter studies.
While our study may not have revealed new insights into the pathological development of these tumors or treatment approaches distinct from previous research, it contributes to a deeper understanding of this disease within the medical literature. The accumulation of data from similar studies provides a foundation for future investigations. We emphasize that a pertinent statistical analysis can and should be undertaken when more cases are reported in the literature, thereby advancing our comprehension and management of the disease.
”Specific comments: Line 88. DBCL should be provided with the full name on the first use. ”
Response: Thank you for your observation. We corrected as follows:
According to the WHO 5th edition of Haematolymphoid Tumors, diffuse large B cell lymphoma (DLBCL) were subclassified into germinal-center B-cell (GCB) and non-GCB subtypes, based on immunohistochemical (IHC) analyses (15) (lines 89-90).
Reviewer 2 Report
Comments and Suggestions for Authors
This is an interesting retrospective analysis of 11 cases of thyroid lymphoma, a quite infrequent thyroid malignancy. I do have some questions, regarding the medical history of these patients, as presented in Table 2
1. It is clear that the diagnosis was retrospective in 11 out of 12 patients. Was it also unsuspected?
2. Do you have any ultrasound image of the thyroid nodules/mass before surgery. Some descriptions of the ultrasound reports are presented, but the images would be much appreciated. It is also not mentioned which was the lymph node status of these patients – were lymphadenopathies observed in thyroid ultrasound before surgery?
3. It is not clear how many patients had positive thyroid antibodies, since in the table, 2 have history of autoimmune thyroiditis (how was the diagnosis established?) but other 2 have positive antibodies. For the rest, thyroid antibodies were never measured? Were they ever evaluated by an endocrinologist before being sent to surgery?
4. Why was surgery the first treatment option for the patient with prior diagnosis of lymphoma?
5. Do you have any comment regarding the number of patients sent to surgery without prior FNA (10 out of 11)?
Comments on the Quality of English LanguageNo comments
Author Response
Thank you for your consideration and the in-depth analysis of our manuscript.
- It is clear that the diagnosis was retrospective in 11 out of 12 patients. Was it also unsuspected?
Response: Thanks to the reviewer for pointing this out.
Only one patient had a suspicion for thyroid lymphoma. For the other ones a lymphoma diagnosis was not suspected being diagnosed after surgery by pathological examination.
- Do you have any ultrasound image of the thyroid nodules/mass before surgery. Some descriptions of the ultrasound reports are presented, but the images would be much appreciated. It is also not mentioned which was the lymph node status of these patients – were lymphadenopathies observed in thyroid ultrasound before surgery?
Response:
Unfortunately, we don’t have images from the ultrasound, neither the status of the regional lymph nodes before the surgery.
- It is not clear how many patients had positive thyroid antibodies, since in the table, 2 have history of autoimmune thyroiditis (how was the diagnosis established?) but other 2 have positive antibodies. For the rest, thyroid antibodies were never measured? Were they ever evaluated by an endocrinologist before being sent to surgery?
Response: Thank you for your observation.
Unfortunately, considering the retrospective nature of the study, some clinical data are not complete. All the clinical data and biochemical analyzes that we had came from the clinical records and operative reports of the Surgery Department and can be found in table 2. It is very possible that the patients were examined by an endocrinologist and were referred to a surgical department, but we do not have this information.
- Why was surgery the first treatment option for the patient with prior diagnosis of lymphoma?
Response: For the patient diagnosed with lymphoma by FNA, it was the decision of the clinician and the surgeon for a surgical treatment. We must specify that the patient was diagnosed in 2011, when the data on the treatment of thyroid lymphoma were limited.
- Do you have any comment regarding the number of patients sent to surgery without prior FNA (10 out of 11)?
Response: Thank you for your observation.
We believe that this approach to a growing mass in the anterior cervical region is due to the lack of a suspicion of lymphoid neoplasia.
Reviewer 3 Report
Comments and Suggestions for Authors
Dear authors,
The manuscript “Primary thyroid lymphoma: a retrospective-observational study in a single institutional center”, medicina-2888641, describes 11 cases of PTL. It is written in a nice and fluent way; it is easy to read and understand. But, as long as PTL is a rare neoplasia of the thyroid gland, the sample size of this study is too small to draw any general conclusions. Furthermore, the study that is being presented in this way adds nothing new to the body of knowledge currently available about PTL, as far as I could notice. Could you kindly describe the key findings or implications of your work, how it differs from earlier research, and how it advances our understanding of PTL? especially considering that the study had numerous limitations, which are clearly noted in the text. Furthermore, it is evident that the group with a median age of 55 would have a higher chance of survival than the group with a median age of 75, hence the conclusion should be revised.
Besides these major remarks, here are some minor:
1. Line 28 in the abstract: I suggest changing the term “good evolution” with something more convenient for neoplasia development.
2. Line 88: The description of the DLBCL acronym is missing.
3. A description of the IHC protocol (or reference) is missing. Please add.
4. The data on the antibodies used is missing. Please add comprehensive information on the antibodies, including dilution for IHC staining and the antibody's manufacturer.
5. The explanation for why the given markers were tested is missing. Please add.
6. I suggest shifting Table 1 into the results section.
7. I suggest you present the statistics of ages via the median value, not the mean.
8. I suggest shifting Table 2 into the M&M section.
9. The legend of Fig.1a, b, and c and Fig2 a,b are missing methodology descriptions (H&E staining). Please add.
10. Line 222: Similarly, as in the abstract, I would recommend some other term instead of “evolution”.
11. Line 322-324: It is written: “Graff-Baker et al. recorded a general survival at 5 years of 66%, depending on histological subtype: 96% for E-MZL lymphoma, 75% for DLBCL, and 87% for FL (2)”. Can you explain how 66% could be the general (mean) survival if particular survivals are 96%, 75% and 87%?
Author Response
Thank you for your consideration and the analysis of our manuscript.
- Line 28 in the abstract: I suggest changing the term “good evolution” with something more convenient for neoplasia development.
Response: Thanks for your suggestion. We have changed the words as follows:
PTL is a lymphoid neoplasia with favorable outcome, with relatively long survival if it is diagnosed at younger ages. (lines 29-30)
We also modified the conclusion as follows (lines 446-458):
Primary thyroid lymphoma is a rare neoplasia, but this tumour must be taken into account, especially in women with a history of autoimmune lymphocytic thyroiditis presenting with a rapidly growing tumour mass in the anterior cervical region, accompanied by local compressive signs. It is a lymphoid neoplasia with a favourable outcome, with a relatively long survival if it is diagnosed at younger ages. These findings need to be confirmed through large-scale multicenter studies.
While our study may not have revealed new insights into the pathological development of these tumors or treatment approaches distinct from previous research, it contributes to a deeper understanding of this disease within the medical literature. The accumulation of data from similar studies provides a foundation for future investigations. We emphasize that a pertinent statistical analysis can and should be undertaken when more cases are reported in the literature, thereby advancing our comprehension and management of the disease.
- Line 88: The description of the DLBCL acronym is missing.
Response: Thank you for your observation. We corrected as follows:
According to the WHO 5th edition of Haematolymphoid Tumors, diffuse large B cell lymphoma (DLBCL) were subclassified into germinal-center B-cell (GCB) and non-GCB subtypes, based on immunohistochemical (IHC) analyses (15) (lines 89-90).
- A description of the IHC protocol (or reference) is missing. Please add.
Response: Thank you for your observation. We added the immunohistochemical method used in our study (lines 100–104 and Table 1).
- The data on the antibodies used is missing. Please add comprehensive information on the antibodies, including dilution for IHC staining and the antibody's manufacturer.
Response: Thank you for your observation. We added the characteristics of antibodies in Table 1.
- The explanation for why the given markers were tested is missing. Please add.
Response:
At the moment of the diagnosis, this markers were the only ones available in our department, and were used to rule out another type or subtype of tumor.
- I suggest shifting Table 1 into the results section.
Response:
We appreciate your suggestion. We have included Table 1 in the Materials and Methods section, which displays the characteristics of the antibodies used and the staining method. Additionally, in the Results section, we have introduced Table 4, showcasing the IHC profile of the tumors.
- I suggest you present the statistics of ages via the median value, not the mean.
Response:
Thank you for your suggestion. For us the mean value is more reliable than the median value, but if necessary, we can change it or add median along with the mean. Thank you!
- I suggest shifting Table 2 into the M&M section.
Response:
With respect to the reviewer’s suggestion, we would like Table 2 to remain in the Results sections, considering that the information contained are the results of clinical and paraclinical investigations.
- The legend of Fig.1a, b, and c and Fig2 a,b are missing methodology descriptions (H&E staining). Please add.
Response: Thank you for your observation.
We added H&E staining in the legend of the figures 1-3.
- Line 222: Similarly, as in the abstract, I would recommend some other term instead of “evolution”.
Response: Thank you for your observation . We modified as follows:
Tratment and survival data (line 287)
We also modified modified the text that is now found in the lines 297-298 as following:
In patients diagnosed with E-MZL lymphomas, we noted the following course of the disease: one patient (13.3%) survived for 9 years, one patient (33.3%) survived less than a year and another patient (33.3%) died 2 years after diagnosis due to an aortic aneurism rupture
- Line 322-324: It is written: “Graff-Baker et al. recorded a general survival at 5 years of 66%, depending on histological subtype: 96% for E-MZL lymphoma, 75% for DLBCL, and 87% for FL (2)”. Can you explain how 66% could be the general (mean) survival if particular survivals are 96%, 75% and 87%?
Response:
I think that general (overall) survival at 5 years was 66% and five-year disease-specific survival was 96% for E-MZL lymphoma, 75% for DLBCL, and 87% for FL.
To clarify the information, we corrected as follows>
Graff-Baker et al. recorded a five-year disease-specific survival, depending on histological subtype, as follows: 96% for E-MZL lymphoma, 75% for DLBCL, and 87% for FL (2). (paragraph 413-415):
Round 2
Reviewer 3 Report
Comments and Suggestions for Authors
Dear authors,
The revised version of the manuscript “Primary thyroid lymphoma: a retrospective-observational study in a single institutional center”, medicina-2888641, has been significantly improved. However, there are still a few minor corrections that should be implemented before it can be published:
1. The presentation of the influence of age on the survival of the patients was not well done. There is no comment concerning my second major suggestion from the previous report: “it is evident that the group with a median age of 55 would have a higher chance of survival than the group with a median age of 75, hence the conclusion should be revised.” At least, the authors should add a brief description of the groups (e.g., insert some text in lines 306-310) and make it obvious that there is a 20-year difference between the median values of tested groups. The authors should be much more careful when summarizing, analyzing, and drawing conclusions from their data.
2. The explanation and justification for testing specified IHC markers should be inserted into the manuscript.
3. The explanation of IHC scoring should be added to the Material and Methods section and briefly in the legend of Table 4. The staining is given as +/ - and in % in Table 4. The authors should uniformize the presentation of the results in Table 4 and set the cut-off value for the positive result.
Author Response
We thank you for your observations and suggestion.
- The presentation of the influence of age on the survival of the patients was not well done. There is no comment concerning my second major suggestion from the previous report: “it is evident that the group with a median age of 55 would have a higher chance of survival than the group with a median age of 75, hence the conclusion should be revised.” At least, the authors should add a brief description of the groups (e.g., insert some text in lines 306-310) and make it obvious that there is a 20-year difference between the median values of tested groups. The authors should be much more careful when summarizing, analyzing, and drawing conclusions from their data.
Response: Thank you for helping us to improve this part. We revised and rephrased some parts of the manuscript to point out the differences between these groups, based on the age at the time of diagnosis.
The changes are:
We identified eight women (72.73%) and three men (27.27%), with a mean age at the moment of diagnosis of 68 years and a median age of 72 years (ranging from 50 to 80 years old).(lines 139-141)
We classified patients into two categories: those aged 70 or younger and those older than 70. The mean age for the first group was 59,6 years, with a median age of 59 years, while the second group had a mean age of 75 years and a median age of 74.5 years. Among the patients in the first group, four out of five (80%) survived more than 5 years, while in the older group, only two out of six (40%) patients survived more than 5 years. (lines 336-340)
In terms of post-diagnosis survival, 60% survived beyond 5 years, and 40% lived over 9 years. Those diagnosed at or below 70 years of age exhibited longer survival, probably due to a multitude of factors such as their robust immune system, tumor characteristics, aggressive treatment approach, early detection, higher treatment adherence, and better response to therapy. (lines 451-454)
It is a lymphoid neoplasia with a favorable outcome, with a relatively long survival if it is diagnosed at younger ages, most likely due to the overall status of the patient. (lines 483-485)
- The explanation and justification for testing specified IHC markers should be inserted into the manuscript.
- The explanation of IHC scoring should be added to the Material and Methods section and briefly in the legend of Table 4. The staining is given as +/ - and in % in Table 4. The authors should uniformize the presentation of the results in Table 4 and set the cut-off value for the positive result.
Response: Thank you for the suggestion. We modified the Material and Methods sections accordingly as following:
„These IHC markers have been used to rule out another type of tumor and to establish the subtype of the lymphomas. At the moment of the diagnosis, these markers were the only ones available in our department.
For the Ki-67 immunohistochemical marker, approximately 500 cells were counted using an eyepiece with a grid in a ×400 magnification, in representative areas which do not to contain residual germinal centers, hot spots of proliferation or proliferating T cells. The Ki-67 index was calculated as the percentage of positive cells by averaging the values obtained for the two areas (count–Ki-67 index) [17]. For all other stains, we classified them as positive if more than 30% of the tumoral cells exhibited the specific antibody, whether cytoplasmic or nuclear, as specified in each stain manufacturer's prospectus. Since there is no established requirement to report the percentage of tumoral cells for these stains, we opted to categorize them simply as positive or negative”. (lines 106-118)
Also, we modified the footnote of Table 4:
„Tumor cells expressing the antibody in ≥30% of cells are considered positive (+), while those exhibiting the antibody in less than 30% of cells are considered negative (-)” (lines 209-211).